# A Dynamic Plane Prediction Method Using the Extended Frame in Smart Dust IoT Environments

**DOI:** 10.3390/s20051364

**Published:** 2020-03-02

**Authors:** Joonsuu Park, KeeHyun Park

**Affiliations:** Department of Computer Engineering, Keimyung University, Deagu 42601, Korea; joonsuupark@kmu.ac.kr

**Keywords:** Internet of Things, machine learning algorithms, learning (artificial intelligence), software defined networking, computer network management, smart dust, dual plane development kit

## Abstract

Internet of Things (IoT) technologies are undeniably already all around us, as we stand at the cusp of the next generation of IoT technologies. Indeed, the next-generation of IoT technologies are evolving before IoT technologies have been fully adopted, and smart dust IoT technology is one such example. The concept of smart dust IoT technology, which features very small devices with low computing power, is a revolutionary and innovative concept that enables many things that were previously unimaginable, but at the same time creates unresolved problems. One of the biggest problems is the bottlenecks in data transmission that can be caused by this large number of devices. The bottleneck problem was solved with the Dual Plane Development Kit (DPDK) architecture. However, the DPDK solution created an unexpected new problem, which is called the mixed packet problem. The mixed packet problem, which occurs when a large number of data packets and control packets mix and change at a rapid rate, can slow a system significantly. In this paper, we propose a dynamic partitioning algorithm that solves the mixed packet problem by physically separating the planes and using a learning algorithm to determine the ratio of separated planes. In addition, we propose a training data model eXtended Permuted Frame (XPF) that innovatively increases the number of training data to reflect the packet characteristics of the system. By solving the mixed packet problem in this way, it was found that the proposed dynamic partitioning algorithm performed about 72% better than the general DPDK environment, and 88% closer to the ideal environment.

## 1. Introduction

The Internet of Things (IoT) involves a close connection between the physical world and the digital world [1,2,3,4,5]. In the Internet of Things (IoT) paradigm, many of the objects that surround us will be connected to a network in one form or another [6]. IoT is already incorporated in many fields (e.g., transportation, smart city, smart home, smart health, smart home, e-governance, assisted living, e-education, retail, logistics, agriculture, automation, industrial manufacturing, and business/process management, etc.) around us, so we are experiencing IoT technologies directly and indirectly [4,6,7,8,9].

In its original concept, smart dust integrated microelectromechanical systems (MEMS) sensors, a semiconductor laser diode and MEMS beam-steering mirror for active optical transmission, a MEMS corner-cube retroreflector for passive optical transmission, an optical receiver, signal-processing and control circuitry, and a power source based on thick-film batteries and solar cells [8]. In a broad sense, smart dust technology, which falls under the category of IoT, is a technology that can measure and manage the surrounding temperature, humidity, acceleration, and pressure, etc. via the wireless network [8,9,10]. Sometimes, a smart dust environment can easily collect information around the devices, but it is deployed in areas that are difficult to access, which makes it challenging to manage and repair [3]. Airborne distribution is a useful means of placing devices in environments that are difficult to access, such as mountains, frozen lands, jungles, etc. However, airborne distribution has the drawback that devices may be densely placed in certain areas, and absent in others. This device crowding caused by airborne distribution can lead to an explosive increase in data in certain areas. In addition, packets related to disconnection or reconnection increase due to problems of devices (low battery, breakdown, and aging), so the number of packets that need to be transmitted increases, and the network performance decreases. In addition, the occurrence of an increased density of nodes can cause bottlenecks, which can lead to very fatal problems that can degrade the whole system throughput as well as black out the entire system [3].

One way to solve the above problem is to apply Intel’s (Santa Clara, CA, USA) Dual Plane Development Kit (DPDK) [11,12] to your system to speed up data processing. This type of approach has been very effective in terms of achieving increased throughput, but resulted in a new issue that reduces the throughput when signal packets and user data packets are mixed in similar proportions, because DPDK does not support dynamic partitioning [13,14,15]. This problem is called the mixed packet problem [3,14]. The problem is even more pronounced when the ratio of packets changes dramatically. In other words, throughput reduction is more pronounced when the ratio is constantly changing with the two types of packets intermingled, and especially when the rates of the two types of packets are similar [14]. In this paper, we propose an algorithm that avoids rather than eliminates the mixed packet problem by estimating the ratio of planes. We also then go a step further and reproduce various situations and compare them with the system with our solution.

The remainder of this paper is structured as follows: Section 2 discusses previous studies in this area. In Section 3, we introduce our experiments and findings to identify the mixed packet problem, and the prediction algorithm for the dynamic plane partition, which is designed to achieve this purpose. Section 4 verifies the validity of the proposed algorithm, and Section 5 provides some conclusions.

## 2. Related Works

In this section, we will provide an overview of DPDK and introduce other studies that have used DPDK to increase throughput.

### 2.1. DPDK (Dual Plane Development Kit)

The DPDK increases packet throughput in the whole system [11,12,13,14,15]. It is a set of libraries and drivers for high-speed packet processing developed by Intel [11,12]. In typical communication, a Network Interface Controller (NIC) requests the Operating System (OS) to process packets, and the OS allocates a CPU to process the packet. In this process, the processing of packets is delayed by OS interrupt, memory copy, and so on. The DPDK is designed to realize high-speed packet processing by excluding OS actions (e.g., interrupt [11,12], memory copy, etc.) [12].

Although the DPDK has the advantage of increasing the packet processing speed of the system, it is difficult for several processes to share one network interface, and the protocol of the system that is already operating needs to be modified according to the DPDK. There is also a disadvantage in that it is necessary to directly create the function corresponding to TCP/IP to increase the performance. Most significantly, the DPDK takes into account the packets of the OS’s waiting queue, but it does not take into account the waiting queues of the NICs, which can cause a number of potential problems.

### 2.2. A High-Performance Implementation of an IoT System Using DPDK

J. Pak and K. Park proposed a method of using the dual plane architecture of data plane and control plane to process various kinds of IoT devices (including smart dust devices) and a large amount of data [15]. As a control plane processes signaling data and a data plane processes user data, it can simultaneously process data on different planes depending on the type of data received. The authors also proposed a method of interworking between two different planes.

The dual plane architecture works very efficiently in an IoT environment that has a typical number of devices, and the throughput is dramatically increased. But, in an IoT that goes beyond the limits of general IoTs (such as a smart dust environment), since the ratio of planes is fixed, throughput may be decreased when the ratio is constantly changing with the two types of packets intermingled. In other words, this suggests that the mixed packet problem described above may occur.

## 3. The Dynamic Plane Partition Algorithm

Section 3 explains the structure of the entire system, the algorithms that partition the plane dynamically and a few environments.

### 3.1. System Overview

This research deals with some elements of the high-speed packet processing system for a smart dust environment in IoT environment. Figure 1 shows an overview of the high-speed packet processing system for a smart dust environment and the part covered by this paper.

The system consists of dust devices, relay dust devices, processing node pool, pool controller, and learning processor. Dust devices are very small sensors that collect information about the environment around them, and which may or may not also have a very small amount of computing power. Relay devices are the same as the dust device, and only add the ability to relay packets to the dust device. That is, the relay device is a dust device capable of transmitting data of a lower node to an upper node of the network. A processing node pool is a group of nodes called processing nodes. These have either a data or a control plane depending on the type of packet that they can process. If a node has control planes, it becomes a control plane node (CP-Node); if the node has data planes, it becomes a data plane node (DP-Node). Their roles are not fixed, and can be switched by the pool controller depending on the ratio of packets in the current system. The pool controller requests the learning processor to predict the ratio of processing nodes that is most suitable for the current system, and manages the ratio of processing nodes according to the predicted ratio. The learning processor that predicts the ratio of CP-Nodes to DP-Nodes is the focus of this paper.

### 3.2. Software Configuration of the System

In this system, device and relay device operate as a simulation. The introduction is omitted because it only serves to relay packets. The core software parts are the Pool Controller part and the Learning Processor part. These receive the packet from the relay devices, create it into a suitable data (extension permuted frame) for the system, utilize it as training data, and finally determine the node that provides the service.

When a packet is received from the outside of the system to the receive module of the pool controller, the packet is immediately delivered to the learning processor. The learning processor collects the received packet using an internal collector module to process the packet into the appropriate data used by the system. The collected packets are made of eXtended Permuted Frame (XPF) to serve as training data. XPF is used as training data to train the network. After that, it returns the *Rd* (the ratio of data plane) that is learned by the module responsible for learning to the pool controller. The pool controller, which grasped the packet trend of the entire system with *Rd*, selects a suitable control-plane node in the system and handles the packet. Figure 2 illustrates this process.

### 3.3. A Prediction Algorithm for the Dynamic Plane

To dynamically control the ratio of the processing nodes, it is necessary to predict the ratio of packets generated in the entire system in the future. The Convolutional Neural Network (CNN) [16] and the Recurrent Neural Network (RNN) are generally known to outperform the ANN. CNN’s convolution operation solves problems such as the disappearance of backpropagation and slowing speed on ANN by leaving the relationship of each input vector or compressing specific information. This means that CNN is designed to be specific to images in which pixels are closely related [17]. For this reason, it is not appropriate to apply simple CNN to packets whose data-specific meaning is more important than the relationship of each data.

The RNN is designed to facilitate the processing of natural language because the location of input data is designed to be meaningful [18]. This would seem to be applicable to this system, which predicts the proportion of packets. However, if we examine the problem in detail, it is apparent that we need to predict a system-wide trend rather than predict the type of the next packet. In addition, for the RNN to be meaningful, there must be a relationship between the current packet and the next packet, but we assume that the packets are independent of each other, so a simple RNN is not suitable for this system. Furthermore, RNN is not suitable for us because the proposed system is not intended to send packets sequentially and has no meaning in the sequence.

Both the CNN and the RNN are designed based on the ANN, and the shallower layers are designed to be less sensitive to data and require less learning time than the CNN or the RNN, provided that there is no risk of backpropagation loss and that the input data is not large. The ANN is used in this system because it is necessary and meaningful results can be expected.

Figure 3 shows the structure of the ANN in this system. We have limited the number of layers to two so that the ANN’s backpropagation values do not disappear. The number of neurons at each layer was limited to 256 in consideration of the fact that our input was limited to 128 bytes.

The training data input to the network is provided in the form of eXtended Permuted Frame (XPF), which will be described later. At this time, the data of each XPF is transformed into one dimension and provided to L1 which has 256 nodes. In the final stage of the learning network, L2 uses the Softmax function [16] to output real values (4 bytes). We also use Adam [19] as an optimizer to train nodes on the network (learning rate = 0.001, epoch = 10).

The size of the XPF given as input is determined by the size of N, calculated from the IV in the step of the XPF generation process. For example, if N is calculated as 100, the size of XPF is 1,000,000 times the packet size (128 bytes). An XPF is provided as a training set, and training data are provided as a PF. That is, a PF is used for each calculation to train the network. An XPF can be said to be one big frame with various training data (PF). When N is 100, the size of the PF is 1,280,000 bytes, as long as they share the same GT. When N is 100, the size of the PF is 1,280,000 bytes. A PF sharing the same GT is treated as one training data. This means that 10,000 packets are treated as one training data. 1,280,000 bytes of data are sequentially given to 256 L1 nodes.

We have dealt with eight 4-byte data in our previous studies [12,14,15] and dealt with network environments using TCP/IP. The system is also designed to run on top of the system from previous studies [15]. The packet of the previous study designed a 32-byte space that can handle 40 bytes of TCP/IP headers, 40 bytes of routing information for indicating internal paths, 4 bytes of packet type information and 32 bytes of sending data (8, 4 bytes). In addition, a total of 128 bytes of packets were designed with an extra 12 bytes of space for ease of expansion by adding data [15].

The data used in the training of the prediction algorithm is provided in a fixed size of 128 bytes and grouped according to parameters. There are several reasons for using special training data:(1)Data of variable size must be fixed to pass through the ANN, resulting in overhead.(2)Data of a fixed size can distinguish the kind of packet by whether the specific byte is set without a full parsing of the packet.(3)Batching multiple pieces of data together rather than processing a single piece of data has an overall processing speed advantage.

In the field of image processing and learning, data permutation is a widely used standard for increasing training data [20]. The eXtended Permuted Frame (XPF) in this paper serves as a data extension in terms of packets. The XPF makes it possible to use the same packet as various types of training data by changing its position in the frame and reusing it in another frame. If you write the system-specified XPF, the packet can be reduced by more than 20%. However, this not only breaks the general-purpose nature of the system, which can handle various data, but also ignores the cost of data compression. In short, we did not compress the data because we thought the loss of the general-purpose nature of the system would outweigh the cost-compression advantage.

Data and control packets are combined with each other to form packets in the system. Since each of them is a single entity, it is difficult (and may be impossible) to predict the ratio and tendency of the data in the system with only one piece of data. Therefore, we use the collected data and control packets to perform data permutation, which permuted and amends the data. This not only solves the problem of tendency (combined pattern) predictions, but also by increasing the training data, you can expect to increase accuracy. As such, we define the following concepts to efficiently extend training data and to better predict packet trends:The Cell is the minimum space to hold a packet.The Permuted Frame (PF) is a set of Cells with a Ground Truth (GT).The eXtended Permuted Frame (XPF) is a set of PFs to contain multiple PF patterns.

In addition, the Permuted Frame step performed prior to entering the training phase proceeds in the order shown in Figure 4.

Before explaining the procedure in detail, we explain the parameters and equations. *Rd* is the ratio of data packets, and *Rc* is the ratio of control packets. These have the same relationship as Equations (1) and (2):(1)Rc=1−Rd
(2)Rd=1−Rc

IV (Initial Value) is the initial value of the procedure for calculating *Rd*. IV is a parameter that significantly affects the accuracy of prediction, the size of XPF, and the learning speed, so it should be chosen carefully. IV is a parameter that affects other parameter calculations and should be given in $10^{N}$ form. In addition, if IV is given in $10^{N}$ form, it is easy to utilize log.

The variable N means the number of cells that PF can have. N cells constitute a PF, and the cells of a PF share the same GT. In other words, PF can be defined as Cells sharing the same GT. If IV is given in $10^{N}$ form, the variable N can be assigned the same value as IV (N is equal to IV). But if you did not stick to the form, then you need to calculate N through the following Equation (3).

The variable N and the variable S can be calculated by simply repeating the calculation when writing a program, but we designed Equations (3) and (4) to help provide a more concise and intuitive understanding. The variable N calculated through Equation (3) is used to determine the number of cells in a PF. The variable S calculated through Equation (4) means the interval step of *Rd* between different PFs. If the size of IV increases, then the size of the variable N increases and the size of the variable S decreases. For example, if IV is set to 90, the variable N is calculated 100 and the variable S as 0.01, and if IV is set to 900, the variable N is calculated as 1000 and the variable S as 0.001. This calculation means that when the IV is set to 900, the memory (ratio gap) is consumed as much as 10 times more (closer) than when IV is 90:(3)$$N=10^{\left\lceil log\left(IV\right)\right\rceil }$$

S means the step of GT to be given to PF. In other words, S is an interval between the *Rd*s each PF has. S is closely related to IV and N, as can be seen in Equation (4).
(4)$$S=10^{-\lceil log\left(IV\right)\rceil }$$

Now that we have introduced the parameters, let us introduce the procedure of the algorithm. The proposed algorithm goes through six steps.

In step 1, the hyper parameter IV is set, and the other parameters (N and S) are calculated from it. When calculating N and S, Equations (3) and (4) are used, as previously described. In step 2, we create a cell, the smallest unit of the proposed algorithm. A cell is made in the same size as a packet because it is created to store a packet. A cell can be recognized as a packet with all bytes set to zero. A cell is an empty packet. In step 3, the cell is duplicated to create a PF. This step simply repeats N times. Since all cells in a PF share the same GT, one GT property is sufficient. The GT is the ratio of the data plane node. In step 4, the GT shared by all cells is initialized to S. After being assigned to a GT, S increases by step (like the first S calculated). The step is assigned the same value as S when S is first calculated. In other words, step is the step by which the ratio increases. In step 5, PF is duplicated N times to create PFs. The reason for the need for PFs is that even within the same ratio, the internal packet configuration can be different. For example, when 1 means data packet and 0 means control packet, 1100 and 0011 look like completely different combinations, but GT in the PF is 0.5 for both. The final step is to duplicate the various combinations of PFs N times to create an XPF. PFs have only one GT because the PFs in the PFs share the same GT. As such, it is necessary to extend the PFs with a combination of PFs composed of various ratios, which is called XPF.

#### Example of XPF Generation Process

To help you understand how XPF is generated, we have prepared an example. Figure 5 shows a PF generated by duplicating a cell and assigned GT. In this example, the hyper parameter IV is set to 10. And, for simplicity the packet size is reduced to 10 bytes (the actual packet size 128 bytes, though).

In the first step, calculate N and S by substituting 10 for IV in Equations (3) and (4). N is 10 (equals $10^{\left\lceil log\left(10\right)\right\rceil }$) and S is 0.1 (equals $10^{-\lceil log\left(10\right)\rceil }$). The 3-parameter calculations for the proposed algorithm were done simply. Immediately after the parameter calculation, a cell that is the same size as a packet is created.

In the third step, a cell is duplicated N times (N = 10) to make a PF. It is important to note that each cell is still empty. Another thing to note is the addition of Attribute GT under PF. PF is assigned a value in S in the fourth step.

In the fifth step, PF is expanded once again by N times. When PF simply expands, it shares the same GT (when implementing the program, GT can be assigned to each PF for convenience). Figure 6 shows the PFs once again expanded N times.

In the sixth step, the PFs are once again expanded to N times eXtended PF (XPF). In this step, the PFs are not only expanded by N times, but the GT of each PF is increased by S steps. That is, S is incremented by S each loop (S = {0.1, 0.2, 0.3, …, 0.8, 0.9, 1.0}). Figure 7 shows the result of step 6.

If each step is successful, then the size of XPF becomes ($\mathrm{Packet}\;\mathrm{Size}\times N^{3}$). In this example, N is 10 and packet size is 10, so the size of XPF is 10,000 bytes. Also, the total number of packets that can be contained in an XPF is 1000 ($N^{3}$). This means that we should have at least 20,000 training packets collected. A GT means an *Rd*. Therefore, a PF with a GT of 1.0 should only be filled with data packets. This means that we should have at least 10,000 data packets collected. In contrast, a PF with a GT of 0.0 should only be filled with control packets. Thus, in conclusion, we need to collect at least 10,000 data packets and 10,000 control packets.

In the last step, we select a packet from candidate packets (collected packets) and fill in the XPF. A PF with a GT of 0.5 is filled with 50% data packets and 50% control packets. Note that the 5:5 ratio data does not always have only one order or combination. There may be different combinations of completely different packets; some may consist of the same packets, but they may be completely different orders. Fill in the rest of XPF in same the way you filled in Figure 8, and you’re done. Figure 8 shows some of the PF s in XPF that have a GT of 0.5.

## 4. Experiment

### 4.1. Experimental Materials

By comparing the three environments and the dynamic partition environment that implemented the proposed algorithm, we verified the validity of the proposed algorithm. A rated static partition environment assumes a very ideal environment. It cannot exist in a real environment, because it assumes that you completely know the pattern of the data being collected. Therefore, in this experiment, we expected that the performance of a dynamic partition environment would be better than that of a static partition environment, and as close as possible to a rated static partition environment.

We believe that in adverse environments, such as deserts and lunar surfaces, climate data may change unpredictably, unlike in normal weather conditions. However, we use general weather data because we cannot currently collect data from adverse environments. In addition, there are some Smart Dust data already collected, but little information has been systematically collected by a reliable organization. Therefore, in this study, we use meteorological data in order to use the vast amounts of data collected by trusted agencies.

We simulated the environment for collecting data through the use of temperature sensors installed in various parts of Korea (Seoul, Daejeon, Daegu, Busan, Gwangju). The data transmitted by each node in the simulation uses the data of the last one year (April 2018~March 2019) provided by the Korea Meteorological Administration (KMA) [20] and Open Data Portal [21], which is converted to a 128-byte fixed length packet. Table 1 below shows samples of data collected from [20,21].

In addition, the hyper parameter IV of a dynamic partition environment is set to 0.01, considering the performance of the experimental specifications. The increase of control packets means that a new session connection is attempted or the nodes requiring signaling information increase (in this case, the data packets are relatively reduced). That is, a reduction in data packets indicates the leaving of a device, and an increase in control packets is an indicator of the reconnection of devices. Therefore, we varied the ratios of packets per interval in the experiment (the ratio of each interval is randomly determined) to reflect the dropout and reconnection of the devices (dust and relay dust dives) in the simulation. To facilitate the viewing of the experimental results, result tables and graphs were arranged in order from low *Rd*.

We have performed experiments that prove a dynamic partition algorithm shows excellent performance, and looked for problems and complementary points. We created XPFs with 10,000 of the pre-collected 20,000 packets (10,000 data packets and 10,000 control packets) and used the remaining 10,000 data packets to verify the performance of the experiment.

Table 2 and Table 3 show the specifications under which the simulation is performed. Table 2 shows the specifications under which the learning algorithm operates in the simulation, and Table 3 is the specification in which the remaining parts are operated. The environments of this experiment are virtual environments, and all conditions except the learning part described above are the same.

### 4.2. Environments

The DPDK system does not support the plane partitioning method [13]. Since there are few similar studies at present, to validate the proposed method we built a system to support the dual partitioning environment, a fixed plane partitioning environment using DPDK.

Previous research has examined the question of how to process control planes and data planes in physically independent processing nodes as a way to improve the system performance [13,14,15]. In particular, [14] shows that the processing performance increases when control planes and data planes exist in independent processing nodes. Therefore, in this paper, we have selected and constructed three architectures that reflected the characteristics of previous studies. The configured environments are as follows:A dual environment that has both data and control plane on one processing nodeA static partition environment in which one processing node has only one plane (data or control plane) at a momentA rated static partition environment that has only one plane (data or control plane) at a moment and the ratio of planes at the same rate as the packets being generated in the system.

Since the environment in which DPDK is typically implemented [13] has both control planes and data planes to each of the processing nodes (servers in Figure 9), one processing node can process both kinds of packets [13]. We named this environment a dual environment (see Figure 9).

In the environment in which a processing node is configured to process only one type of data or control packet, only one type of plane exists in the processing node [14]. We named this environment a static partition environment (see Figure 10). In a static partitioning environment, it has a fixed to the ratio of the specified plane. Thus, if packets are received at the same ratio as the system’s plane ratio, they can perform as well as in an ideal environment as if future predictions are certain. However, performance is low in other situations except for certain situations in which the two ratios are coincidentally matched [13].

The environment in which the processing node processes only one type of packet and where the ratio of the packet rate to the processing node is very ideal, we named a rated static partition environment (see Figure 11). A rated static partition environment is not only an environment that assumes an ideal environment, but is also an environment that cannot be implemented in reality, because it cannot accurately determine the ratio of packets in continuous time. This ideal environment is the environment that it is hoped the performance of the proposed environment (the prediction algorithm for the dynamic plane) can approach.

### 4.3. Experimental Results

Table 4 shows the number of processing bits per second when each system is applied. In the following tables and graphs, *Rd* is the ratio of data planes, which must be multiplied by 0.1 to get the same value as described above.

The results of the experiment can be summarized as follows:(1)The dynamic partition environment shows about 88% the performance of the rated static partition environment (12% lower than the rated static partition environment).(2)The dynamic partition environment shows about 16% higher performance compared to the static partition environment.(3)The dynamic partition environment shows about 72% higher performance compared to the dual environment.

#### 4.3.1. Comparison with the Dual Environment

The performance of the dynamic partition environment shows about a 72% increase compared to the performance of the dual environment. The performance of the dual environment is not as good as the performance of the dynamic partition environment due to data bottlenecks (see Figure 12).

In the dual environment, control packets simultaneously received from multiple devices are requested to be processed by the processing node with the shortest waiting queue in the system. Since the control packets can be processed much faster than the data packets (because of DPDK), it seems unlikely to be a problem initially. However, when there is a large number of devices that are intermittently disconnected, the problem faces a new phase. The NIC’s waiting queue is full of control packets when devices are reconnected. At this time, devices that are not disconnected still send data packets to the processing node. Processing nodes trying to process control packets have difficulty accommodating data packets. This means that the mixed packet problem has occurred (see Table 4 on sudden performance degradation when *Rd* is 5). The dynamic partition environment is robust against mixed packet problems (see row *Rd* 5 in Table 4).

Another indicator worthy of attention is the standard deviation. A bigger standard deviation means that there is a large performance difference depending on the type of packet. The dual environment has a standard deviation of about 218,451 and the dynamic partition environment has a standard deviation of about 136,229. When all of this is considered, it is clear that the dynamic partition environment performs better than the dual environment in almost all respects.

#### 4.3.2. Comparison with the Static Partition Environment

The static partition environment solves this problem well, as it is a method designed to solve the mixed packet problem of the dual environment. However, the static partition environment poses the opposite problem to the dual environment. The performance issue related to extreme bottlenecks (see if *Rd* = 5 in Table 4) is resolved, but the performance of the remaining area is degraded. If *Rd* is 1, when there is a sudden flood of control packets, most of the control packets cannot be processed, so another bottleneck results (due to the lack of flexibility of the static partition environment). As can be seen in Figure 13 below, when *Rd* is set to 5, the performance of *Rd* at the pole (if *Rd* is 1 or 9) is degraded.

The standard deviation of a static partition environment is approximately 957,445, which is a larger performance interval than the dual environment and is a difference of more than 4 times from 218, 451, the standard deviation of the dynamic partition environment. Therefore, the dynamic partition environment performs better than the static partition environment in almost all respects.

#### 4.3.3. Comparison with the Rated Static Partition Environment 

As noted above, the rated static partition environment is capable of getting as close to the performance of the dynamic partition environment as possible. We do not want the performance of the dynamic partition environment to exceed the performance of the rated static partition environment, but we expect to see near equivalent performance. Our first goal was to achieve 90% of the performance of the rated static environment (see Figure 14).

Ultimately, we did not reach the target of 90%, but on average reached 88% (see Figure 14 and Figure 15). The standard deviation showed the remaining problem to be solved. The standard deviation of the rated static partition environment is 43,391.84, which is 1/3 that of the dynamic partition environment. The dynamic partition environment requires time to judge the pattern and tendency of the data, resulting in a difference in performance. That is, the above-described performance difference can be considered as a cost required for prediction.

## 5. Conclusions

We have proposed a partitioning algorithm that can predict and process based on the ratio of packets occurring in the system to solve the mixed packet problem, which is a potential problem in the dual environment. In this process, when a general packet is given as training data, it is difficult to predict a ratio from the training data. As a solution, we designed a training data frame called the XPF to predict the proportion of the total data. That is, the XPF was used as a means to solve the mixed packet problem occurring in the dual environment. In addition, we introduced XPF, which transforms the discrete nature of packets into a collective property called ratio, and uses more of the acquired training data.

Using the proposed algorithm, we can improve the performance of the dual plane architecture in a smart dust environment where many devices are connected to generate a large amount of data.

We conducted comparative experiments with various system environments to validate the prediction algorithm through XPF. The performance of the proposed algorithm increases by about 72% compared to the dual environment that includes only DPDK. However, due to the cost of additional calculations, its performance dropped about 12% compared to the ideal environment that existed only theoretically. The evidence from the experiments highlights the importance of the computational cost needed to follow the ideal environment. Currently, the proposed system uses the ANN (Artificial Neural Network) as a predictive model. It is expected to offer an advantage in terms of performance, but considering the speed of learning and the cost of computation, it is hard to think of it as an optimal model. Therefore, further research should be carried out on predictive models and algorithms that can reduce computations and ensure certain levels of accuracy.

## Figures and Tables

**Figure 1 sensors-20-01364-f001:**
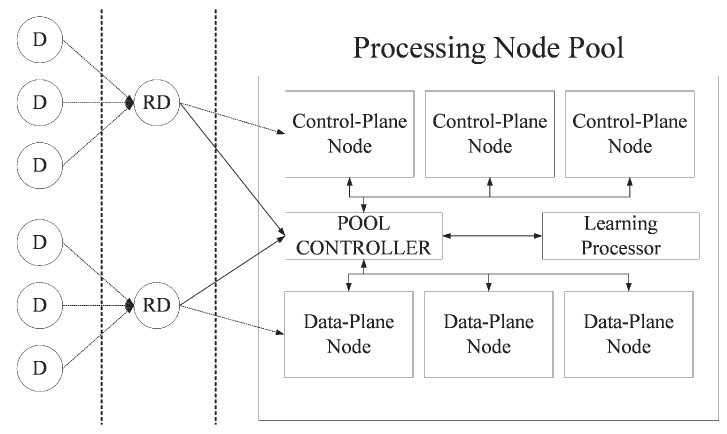
System overview.

**Figure 2 sensors-20-01364-f002:**
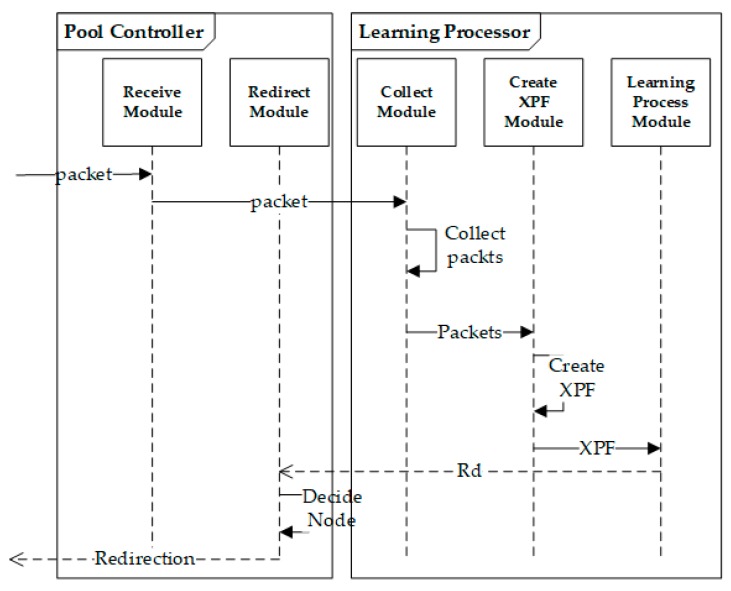
Learning software flowchart.

**Figure 3 sensors-20-01364-f003:**
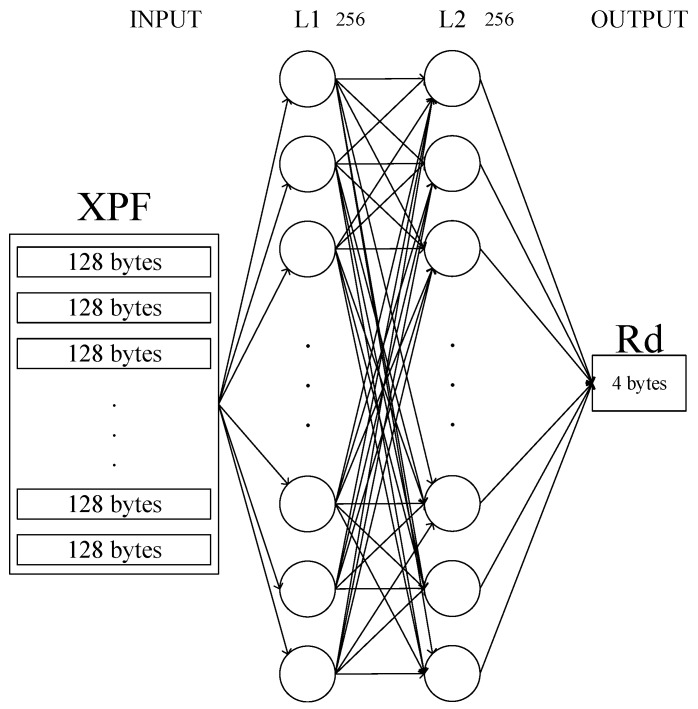
Model structure of the ANN used in the proposed system.

**Figure 4 sensors-20-01364-f004:**
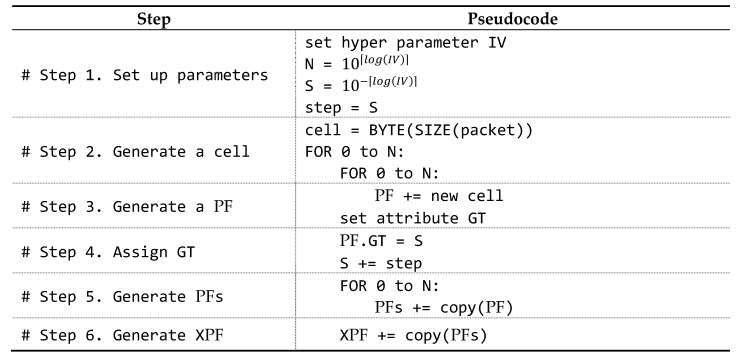
Algorithm of the XPF creation procedure.

**Figure 5 sensors-20-01364-f005:**
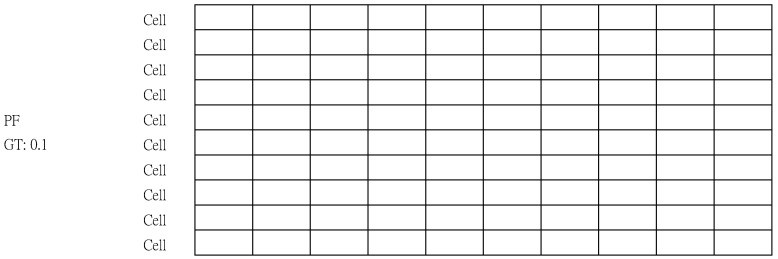
Generated a PF by duplicating a cell (PacketSize×N).

**Figure 6 sensors-20-01364-f006:**
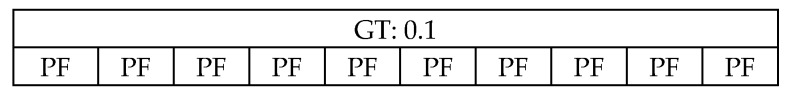
The PFs once again expanded N times (PacketSize×N×N).

**Figure 7 sensors-20-01364-f007:**
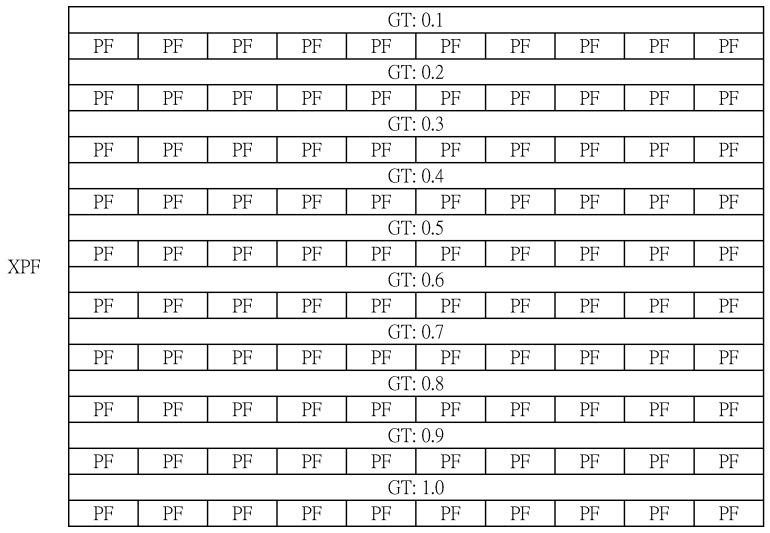
The XPF in the empty state (PacketSize×N×N×N).

**Figure 8 sensors-20-01364-f008:**
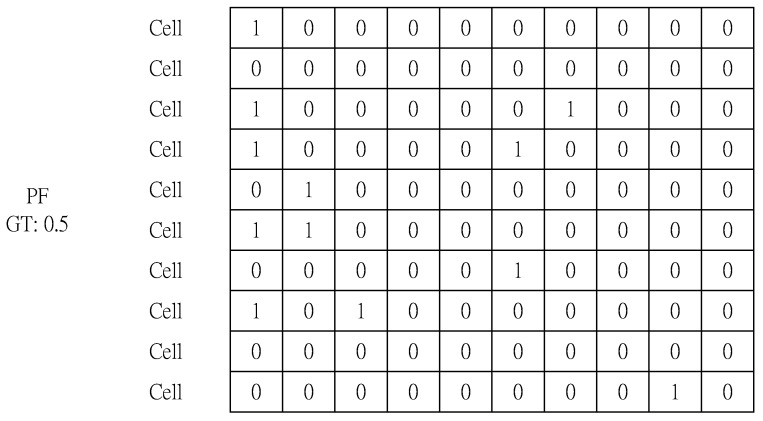
The first PF in XPF that has a GT of 0.5 (A packet whose first byte in a cell is 0 (1) is a control data packet).

**Figure 9 sensors-20-01364-f009:**
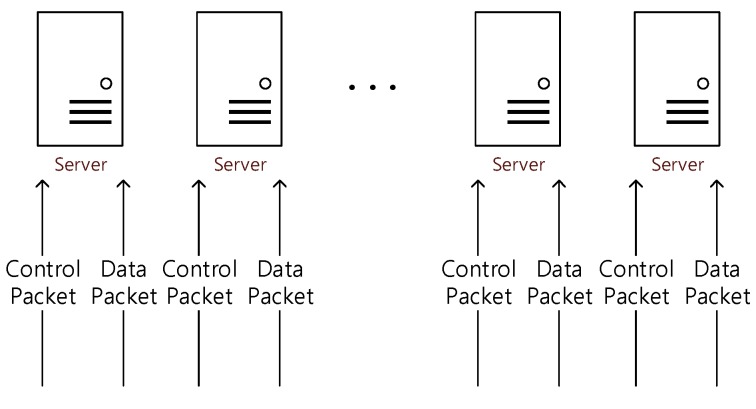
Example of a dual environment.

**Figure 10 sensors-20-01364-f010:**
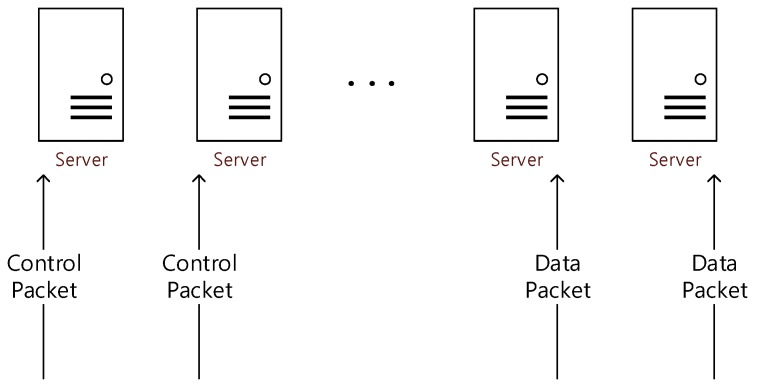
Example of a static partition environment.

**Figure 11 sensors-20-01364-f011:**
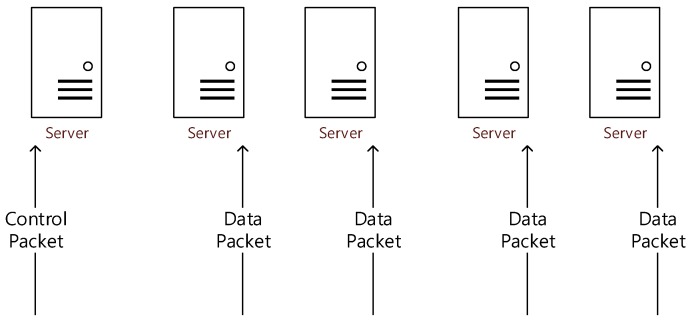
Example of a rated static partition environment by a 2:8 ratios.

**Figure 12 sensors-20-01364-f012:**
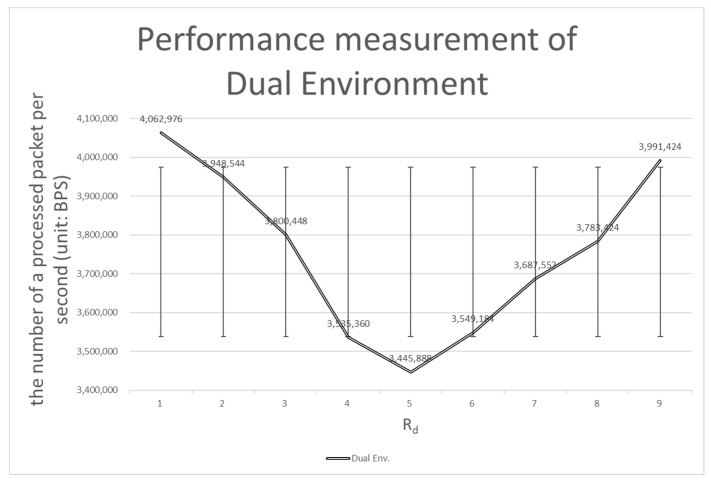
Performance measurement of the dual environment.

**Figure 13 sensors-20-01364-f013:**
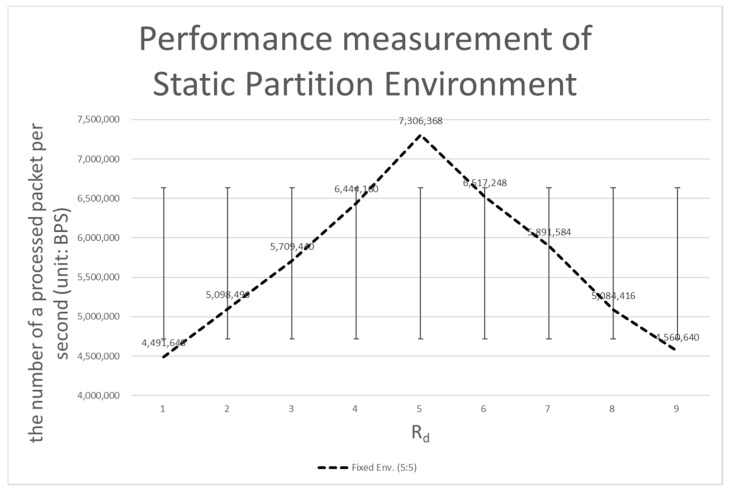
Performance measurement of a static partition environment.

**Figure 14 sensors-20-01364-f014:**
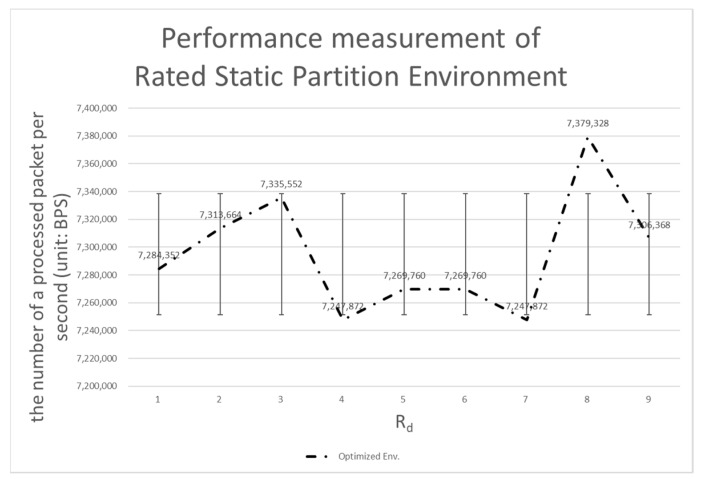
Performance measurement of the rated static partition environment.

**Figure 15 sensors-20-01364-f015:**
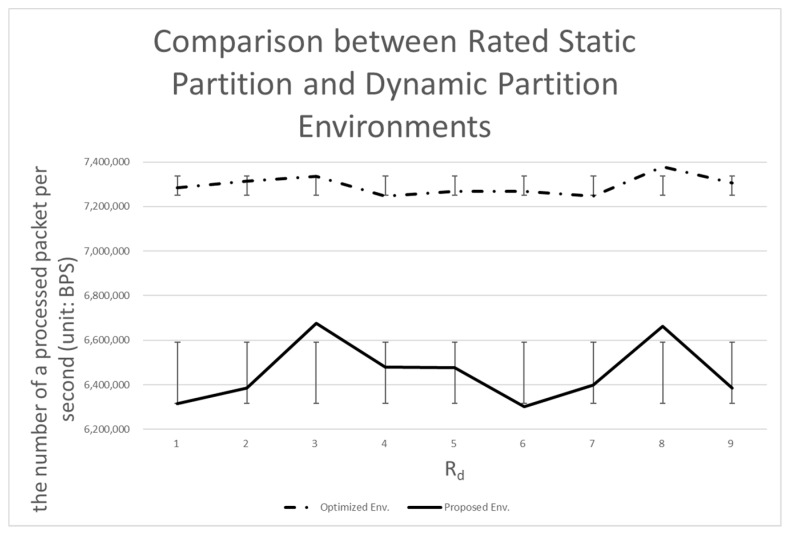
Comparison between a rated static partition and the dynamic partition environment.

**Table 1 sensors-20-01364-t001:** Training data sample.

Observation Point	Time	Temperature (°C)	Humidity (%)	Vapor Pressure (hPa)	Dew Point Temperature (°C)	Sunshine (h)	Solar Radiation (MJ/m2)	Ground Temperature (°C)
Daegu(143)	25 January 2020 8:00	6	77	7.2	2.2	0	0.03	3.9

**Table 2 sensors-20-01364-t002:** Specifications under which the learning algorithm operates.

Component	Item
CPU	i7-4770 3.4GHz
Main memory	16.0 Gb
GPU	NVIDIA GeForce GTX 1050 3Gb X 2
OS	Ubuntu 16.04

**Table 3 sensors-20-01364-t003:** Specifications under which the remaining parts operate.

Component	Item
CPU	i7-4770 3.4GHz
Main memory	32.0 Gb
GPU	NVIDIA GeForce GTX 1050 3Gb
OS	VM Ware (Ubuntu 16.04)
	Docker(Ubuntu 16.04)

**Table 4 sensors-20-01364-t004:** Experimental results of each Environment according to the ratio of packets (size of one packet: 128 bytes, unit: BPS).

*Rd* (%)	Dual Env.	A Static Partitioning Env. (5:5)	A Rated Static Partitioning Env.	A Dynamic Partitioning Env.
10	4,062,976	4,491,648	7,284,352	6,315,520
20	3,948,544	5,098,496	7,313,664	6,384,768
30	3,800,448	5,709,440	7,335,552	6,675,328
40	3,535,360	6,444,160	7,247,872	6,479,488
50	3,445,888	7,306,368	7,269,760	6,477,312
60	3,549,184	6,517,248	7,269,760	6,302,848
70	3,687,552	5,891,584	7,247,872	6,399,744
80	3,783,424	5,084,416	7,379,328	6,663,424
90	3,991,424	4,560,640	7,306,368	6,385,664

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
