# Peer review of "A Dynamic Plane Prediction Method Using the Extended Frame in Smart Dust IoT Environments"

_sensors, 2020, doi:10.3390/s20051364_

Round 1

Reviewer 1 Report

This paper proposes the use of dynamic partitioning algorithm to improve the performance of the general Dual Plane Development Kit introduced by Intel [19]. They focus their contribution on the prediction of the ration of CP-Nodes to DP-Nodes. Based on a training data model based on eXtended Augmented Frame (XAF), they evaluate the performance of their dynamic approach with three other environments : dual environment where each processing node treat both data and control plane; static environment where each processing node treat only one plane; rated static partition environment where the ratio of planes follows the same rate as the generation of the packets. They claim that their approach outperforms the standard approach by 72% and it achieves 88% of the performance of the rated static partition environment (which is considered the optimal case).

Strong Aspects:

Smart Dust IoT is one of the main challenges and all contributions to reduce computational power is beneficial for their development. The proposal to rationalize the methodology to dynamically distributed computational resources to reduce data queuing and cope with unbalanced mixed packed problem is very interesting.

Weak Aspects and Suggestions:

It is defined on Section 2.1 that DPDK is designed to perform high-speed packet processing by excluding OS interruptions, but no explanation of such interruptions is detailed later on the paper. Authors affirm on Section 3.3 that CNNs and RNNs are not suitable to their problem, but actually their conclusion is based on the chosen approach to build their leaning processor. Other approaches could convey with CNNs and RNNs. Authors do not justify why they choose their training data with 128 bytes and what is the impact of their choice. Furthermore, the batching of multiple pieces of data has an overall processing speed advantage, as claimed, but they do not explain how this shall impact the dual environment. Authors claim that iRd parameter significantly affects the accuracy of prediction and the learning speed, and it should be chosen carefully, but they do not present any result to justify their affirmations. On section 4.1, they varied the ratios of packets per interval in the experiment, but no detail about how the randomness was introduced. The randomness can have a strong impact on the performance of different environments.

Author Response

This paper is the extended version of the paper presented in IEEE IEMCON2019 (Vancouver).

We would like to express our sincere gratitude to the Editor-in-Chief, the guest editors, and the reviewer for their insightful comments and suggestions. These comments have helped us to further improve the paper. We believe that the revised version addresses reviewer all comments.

Reviewer 2 Report

The paper proposes a dynamic partitioning strategy to resolve the mixed packet problem in DPDK, which can result in throughput reduction.

In summary, it is not a high quality paper and should not be accepted.

The following are the detailed comments:

Strengths:

The idea of dynamic partitioning is interesting and worth studying. The system overview, software configuration of the system and example is clearly written. The analysis of the experimental results is detailed.

Weaknesses:

The authors need to improve their English writing ability considerably.  The sentences in this paper is verbose, adjacent sentences often express the same meaning, such as the second sentence on line 79 and the second sentence on line 80, the second sentence on line 194 and the second sentence on line 195. There were some editing errors, line 174 is a table or an algorithm not a figure, line 186 should be 10^n instead of 10n, the word ‘equation’ should not be used in the plural in  line 187, line 241 has a reference error, line 305 is missing a word ‘rated’, the variable to calculate N and S in the pseudocode step 1 should be iRd but not x and so on. These problems can avoidable after careful examination. In the algorithm description, the relationship among N, S and iRd is not clear the physical meaning of S is ambiguity. If the equation (3) and equation (4) were designed by you, please explain the reason for the design, otherwise please quote the source. A key question is what are the challenges of your proposal? In other words, why is it necessary to predict the division of data plane nodes and control plane nodes given the ratio of data packet?  Can't one just use Rd as a partition ratio? What are the advantages of artificial neural networks that make you choose it? I only see the shortcomings of ANN compared to CNN and RNN in your article and there are no references to neural networks. Besides, the configuration of neural network, such as the number of layers and the number of neurons in each layer, was not explained. The scheme is a little too direct. The input of the neural network is a large-scale sparse 4D tensor resulting in a mountain of parameters and high running overhead. Since the partitioning strategy has nothing to do with the location of packets and control packets, but only depends on the number, why not compress the input of neural network? Only one metric (the number of packages) is compared in the experiment, and the meaning of the numerical values in the table 3 is not even shown. In the case of Rd = 50%, the performance of the static dynamic partitioning is better than the static partitioning, so the proposed scheme is not effectively solve the problem of the throughput reducing when signal packets and user data packets are mixed in similar proportions.

Author Response

(The authors gave the same response as above.)

Reviewer 3 Report

The paper "A Dynamic Plane Prediction Method using the Extended Augmented Frame in Smart Dust IoT Environments" presets an approach for dynamic partitioning of communicating nodes on planes based on an ANN prediction. The approach is designed for an IoT smart dust environment. It is build on top of the Intel Dual Plane Development Kit architecture. It aims to solve the communication performance degradation due areas of high density of nodes in case of random deployment.

The introduction is well written and provides a good description of the problem. The related work section provides a very brief review on the work related to DPDK.

The main authors contribution is presented in Section 3.3. They uses a two layers ANN to predict the ratio of packets generated in the entire system. The design of this part is very poor presented. Moreover, the learning algorithm used to train the ANN is just briefly described. The reader can guess a kind of supervised learning from description. However, it is very unclear how the training data is designed and collected for the training process.

The expecting output vectors of training patterns are not presented at all. There are many details of the model which are not presented as for example the cost function, the learning rate, the optimizer used or the number of epochs. Even the architecture of the network used is not very clear (e.g. from Fig 3, which suggest a MLP architecture, it is not clear how the 128 bytes of the input vector are distributed to the 256 nodes of the first layer, or how all the 256 values resulting from the second layer are combined to obtain the 4 Bytes on the output).

The physical meaning of the data augmentation process is not explained (e.g. in case of training data for image recognition, the variation of colors in a certain interval corresponds to analogue variation on real picture colors due illumination, reflections or other factors. The interval itself is determined based on this variation.).  

Some variables used in designing the algorithm presented in Fig. 4 are confusing. The authors defines the hyper parameter iRd as [211] "The hyper parameter iRd is the initial value given to Rd, and only one is given per learning module." and give example of [225] "iRd is set to 90" or [226] "iRd is set to 900". However, the Rd variable, which is [209] " the ratio of data packets" should be a number in the [0.0-1.0] interval, also according to eq. (1) and (2). Therefore it is unclear how the iRd, which is the initial value given to Rd, could be greater than 1.0 (90 or 900!).

The validation section does not present any comparison with other algorithms or methods used for the same purpose. It is also unclear how the data used for validation, which represents meteorological temperatures in various parts of Korea, are relevant for this algorithm (the weather evolve very slowly, therefore no communication issues should appear when the measures should be transmitted at large time intervals as minutes of tents of minutes).

Author Response

First of all, we would like to express our sincere gratitude to the Editor-in-Chief, the guest editors, and the reviewers for their insightful comments and suggestions. These comments have helped us to further improve the paper. We believe that the revised version addresses all comments from the reviewers.

Reviewer #3

They uses a two layers ANN to predict the ratio of packets generated in the entire system. The design of this part is very poor presented. Moreover, the learning algorithm used to train the ANN is just briefly described. The reader can guess a kind of supervised learning from description. However, it is very unclear how the training data is designed and collected for the training process.

Reply:

   To explain the training data further, we added the source weather data format (including Table 1).

   ==>  See: lines 309 to 312 (no-tracing-version)

The expecting output vectors of training patterns are not presented at all. There are many details of the model which are not presented as for example the cost function, the learning rate, the optimizer used or the number of epochs. Even the architecture of the network used is not very clear (e.g. from Fig 3, which suggest a MLP architecture, it is not clear how the 128 bytes of the input vector are distributed to the 256 nodes of the first layer, or how all the 256 values resulting from the second layer are combined to obtain the 4 Bytes on the output).

Reply:

    To help readers understand it more clearly, we added an explanation of the details of the model. In addition, the learning network (Fig. 3), L2 uses Softmax function [18] to output real values (4 bytes). We also use Adam [21] as an optimizer to train nodes on the network (learning rate = 0.001, epoch = 10).

   ==>  See: lines 158 to 176 (no-tracing-version)

The physical meaning of the data augmentation process is not explained (e.g. in case of training data for image recognition, the variation of colors in a certain interval corresponds to analogue variation on real picture colors due illumination, reflections or other factors. the interval itself is determined based on this variation). Some variables used in designing the algorithm presented in Fig. 4 are confusing. The authors define the hyper parameter iRd as [211] "The hyper parameter iRd is the initial value given to Rd, and only one is given per learning module." and give example of [225] "iRd is set to 90" or [226] "iRd is set to 900". However, the Rd variable, which is [209] " the ratio of data packets" should be a number in the [0.0-1.0] interval, also according to eq. (1) and (2). Therefore, it is unclear how the iRd, which is the initial value given to Rd, could be greater than 1.0 (90 or 900!).

Reply:

    First of all, thank you very much for your review comment. The comment is very valuable. The variable ‘iRd’ is not the initial value of the variable ‘Rd,’ but the seed value used in the process of calculating the variable ‘Rd.’ Therefore, the term ‘iRd’ seems to be inappropriate.

   We also think your opinion about augmentation is reasonable. The training data is permuted, not augmented. Therefore, we have changed the related terms (augmented, AF, XAF, iRd, etc.) throughout the paper. In particular, the variable ‘iRd’ you pointed out was changed to the variable ‘IV’ with the meaning of the initial value of the procedure, and the word ‘augmentation’ was changed to ‘permutation.’

   ==> See: line 218 to 221, 255 to 263, and many places in the paper (no-tracing-version)

The validation section does not present any comparison with other algorithms or methods used for the same purpose.

Reply:

    To the best of our knowledge, few studies have proposed a plane prediction method using DPDK as of now. In addition, the DPDK system does not provide the plane partitioning method that this study proposes. We added some statements to explain this.

   ==> See: lines 331 to 333 (no-tracing-version)

It is also unclear how the data used for validation, which represents meteorological temperatures in various parts of Korea, are relevant for this algorithm (the weather evolve very slowly, therefore no communication issues should appear when the measures should be transmitted at large time intervals as minutes of tents of minutes).

Reply:

   We believe that in extreme environments climate data can change unpredictably, unlike typical weather environments. For example, deserts or moon surfaces have very large daily temperature ranges, unlike normal environments. However, we currently cannot collect climate data from such environments, and therefore we used general weather data instead. We added some statements to explain this.

   ==> See: lines 299 to 304 (no-tracing-version)

Round 2

Reviewer 3 Report

The revised version of the paper "A Dynamic Plane Prediction Method using the Extended Augmented Frame in Smart Dust IoT Environments" clarifies most of the aspects related to the ANN design.  The authors succeed to improve the explanation of the data preparation process. However, there are still some unclear aspects related with the relevance of the data permutation step and its impact on leaning. Finally, the decision of choosing the validation environment is better motivated.